

# Application of relay puncture technique in treating patients with complicated lower extremity arterial diseases

Chengzhi Li[1,*], Huimin You[2,3,*], Hong Zhang[1], Yulong Liu[1], Wanghai Li[1], Xiaobai Wang[1] and Yan Zhang[1]

[1] Department of Interventional Radiology and Vascular Surgery, The First Affiliated Hospital of Jinan University, Guangzhou, Guangdong, China
[2] Department of Endocrinology, The Fifth Affiliated Hospital of Guangzhou Medical University, Guangzhou, Guangdong, China
[3] Medical Imaging Center, The First Affiliated Hospital of Jinan University, Guangzhou, Guangdong, China
* These authors contributed equally to this work.

Corresponding author
Yan Zhang, dsazy@163.com

## ABSTRACT

**Objective:** This study aimed to introduce and evaluate the safety and efficacy of the relay puncture technique in patients with complicated lower extremity arterial diseases.

**Methods:** A total of 21 patients (16 male and five female patients; median age: 68.5 years old), who had suffered from lower extremity arterial diseases between December 2014 and July 2017, were retrospectively collected. For all patients, the contralateral femoral artery was not available for puncture access, and the length of the devices was too short for the brachial artery approach. Therefore, the relay puncture technique, in which the first puncture was performed on the brachial artery, followed by an antegrade puncture on the femoral artery, was used to accomplish the endovascular therapy. Percutaneous transluminal angioplasty and/or percutaneous transluminal stenting were/was used to assess the efficacy of the relay puncture technique. The ankle–brachial index (ABI) and Rutherford clinical classification were used to evaluate the improvement of symptoms after treatment. Patients were followed up for 1, 3, 6, and 12 months, and annually (mean: 16.6 months) after discharge.

**Results:** The relay puncture treatment had a 100% technical success rate, and immediately decreased the ischemic symptoms of patients after the procedure. The ABI significantly increased from $0.33 \pm 0.18$ to $0.75 \pm 0.21$ at the 1-year follow-up time point ($P < 0.05$). No serious complications occurred during the follow-up period. The 1-year primary patency rate was 71.43%.

**Conclusion:** The relay puncture technique is a feasible technique in the hands of experienced and skilled equipment operators for the treatment of lower extremity arterial diseases, when the contralateral femoral artery is not available for puncture, and the length of the device is too short to treat the distal lesion of the femoral artery and popliteal artery through the brachial artery approach.

## INTRODUCTION

Advances in endovascular technologies in recent years have offered more options for patients with lower extremity arterial diseases and expanded the indications for endovascular therapy (*Goodney et al., 2009*). However, in some complex situations when the contralateral femoral artery cannot be punctured, such as when cover stents are implanted on the entire contralateral femoral artery, or the length of the device is too short to treat the distal end of the femoral and popliteal artery through the brachial artery approach, a simple endovascular technique cannot solve the problem (*Yilmaz et al., 2003*). For these situations, a hybrid procedure, such as endovascular and open surgery, is usually an appropriate choice (*Antoniou et al., 2009*). However, some disadvantages of the hybrid procedure include more complicated surgery requiring skilled clinicians to achieve appreciable outcomes, and the requirement of an operating room for performing this procedure (*Murakami, 2018*). Therefore, it is imperative to determine new techniques that can replace hybrid surgery with acceptable efficacy, safety, and simplicity in the treatment of complicated lower extremity diseases.

In the present study, a simple novel endovascular technique, which was named, the relay puncture technique, was employed, in which the first puncture was performed on the brachial artery, followed by an anterograde puncture of the femoral artery. The investigators were able to treat the femoral artery lesions through the brachial artery, while the investigators were able to treat lesions distal to the popliteal artery through the femoral artery. The relay puncture technique was used to treat patients whose contralateral access was not available, since these patients had ipsilateral or contralateral iliac and common femoral axis stenosis or occlusions, a very angulated aortic bifurcation of the iliac, a puncture site with heavy calcifications or a stent, infection of the groin, and a crossover maneuver in the setting of the bifurcated aortic graft or pre-existing iliac kissing stents. The safety and efficacy of the relay puncture technique were evaluated in the present study.

## MATERIALS AND METHODS

### Eligibility

A total of 21 patients (16 male and five female patients), who were within 48–81 years old (median age: 68.5 years old) and suffered from lower extremity arterial disease, were collected in our hospital between December 2014 and July 2017. The diagnosis of each patient was confirmed by ultrasonography and computed tomography angiography (CTA) before the treatment. All patients had clinical symptoms of critical chronic lower extremity ischemia. The ankle–brachial index (ABI) and Rutherford clinical classification (RCC) before and after the operation were used to evaluate the improvement of the lower extremity ischemia of each patient after the procedure. According to the TASC classification, all patients in the present study were type D, and all patients declined to have open surgery, which was probably due to fear of major trauma, advanced age, and/or their family's wishes.

## Institutional review

The experimental protocol of the present study complied with the principles outlined in the Declaration of Helsinki, and was approved by the Ethics Committee of our institution. All subjects provided a signed informed consent.

## Procedures of the relay puncture technique

The patient was placed in the supine position and hypodermically administered with heparin (5,000 units) after a 5-French sheath (Terumo, Tokyo, Japan or Cook, Bloomington, IN, USA) was inserted through the left brachial artery, followed by repeated heparin administration in prolonged procedures. After endovascular treatment of the iliac, femoral, or popliteal artery, the 5F sheath was changed to a 6F sheath, which is a 90-cm-long armored introducer sheath (COOK, Bloomington, IN, USA), in order to achieve adequate support and strengthen the control of the catheter. Then, a 0.035-inch J-type guidewire (Terumo, Tokyo, Japan) or 0.018-inch guidewire (V18; Boston Scientific, Marlborough, MA, USA) supported by a 5F vertebral artery angiography catheter (Cordis, Johnson & Johnson, New Brunswick, NJ, USA) was navigated in the proximal common femoral artery (CFA).

In the present study, the relay puncture technique was used on the ipsilateral femoral artery in the following situations: the contralateral femoral artery could not be punctured (the stents were implanted in the entire contralateral femoral artery in seven patients); the stents were implanted into the aortic artery in the iliac lesion (four patients); there was abdominal aortic aneurysm after the endovascular repair (five patients), which made it difficult for the devices to cross over the bifurcation; the amputation of the contralateral leg did not allow the on-site puncture of the femoral artery (five patients). These five patients only required the recanalization of the contralateral lower extremity artery due to cost problems, and the length of the devices was too short to treat the distal end of the femoral artery and popliteal artery through the brachial artery approach. The lesions in all these patients involved the area of the upper inguinal, through which an antegrade femoral puncture could not be successfully performed.

The detailed procedures of the puncture technique are described, as follows. After the recanalization of the proximal part of the femoral artery, the wire was left in place as a marker. Then, an antegrade puncture was performed at the groin under fluoroscopic guidance, and a 5F 10-cm sheath was inserted. Then, another J-type 0.035-inch guide wire or V18 was used to cross the femoropopliteal lesion, including the tibial disease, followed by a six to eight mm diameter classical percutaneous transluminal angioplasty (PTA) balloon (Inpact, Medtronic, Minneapolis, MN, USA) in the iliofemoral segment, a five to six mm diameter balloon in the femoropopliteal, and a two to three mm diameter balloon in arteries below the knee, and stenting. The balloon inflation times varied within 60–300 s at nominal pressure. Subintimal angioplasty was also used, and subintimal arterial flossing with the antegrade-retrograde intervention (SAFARI) technique was used for reentry. Immediately after the vascularization procedure, the sheath was removed from the femoral artery, and a 6-French vascular closer (Exoseal, Cordis; Johnson & Johnson, New Brunswick, NJ, USA) was used to stop the bleeding of the puncture site.

The median compression time was 5 min (range: 3–10 min). The angioplasty inside the punctured site with a six-mm balloon was helpful in stopping the bleeding after the sheath was removed (6/16). Finally, PTA and percutaneous transluminal stenting were performed on the proximal part of the femoral artery and iliac artery through the left brachial artery approach without any open surgery. After the completion of all these procedures, the sheath in the left brachial artery was withdrawn. All the above-mentioned endovascular procedures were performed by experienced interventional radiologists in our center.

### Endpoints and follow up

The primary endpoint was the efficacy of the relay puncture technique, as indicated by the acute technical success and post-procedure clinical improvement of critical limb ischemia symptoms. The secondary endpoint was the safety of this technique, as indicated by the number or severity of complications associated with this procedure.

Color Doppler ultrasonography and/or CT angiography were performed at 1, 3, 6, and 12 months, and annually, or when symptoms recurred after discharge. Antiplatelet therapy was administered during the whole life of each patient after treatment to prevent restenosis.

### Statistical analysis

Data were presented as mean ± standard deviation. All statistical analyses were performed using SPSS software version 22.0 for PC (SPSS Inc., Chicago, IL, USA). Least significant difference $t$-test was used to compare the ABI before and after treatment. A $P$-value < 0.05 was considered statistically significant.

## RESULTS

### Short term outcomes

A total of 21 patients (16 male and five female patients; median age: 70 years old), who suffered from lower extremity arterial diseases from December 2014 to July 2017, were retrospectively collected. All patients were diagnosed with lower extremity arterial disease before treatment. Preoperative CTA clearly revealed the occlusive lesions of the lower extremity arteries (Figs. 1 and 2). For all these patients, the contralateral femoral artery was not available for puncture access, and the length of the devices was too short for the brachial artery approach. Among these patients, seven patients had cover stents on the entire contralateral femoral artery, four patients had kissing stents in the aortoiliac lesion, five patients experienced endovascular repair of abdominal aortic aneurysm, and five patients underwent amputation of the contralateral leg due to ischemia. After a detailed preoperative discussion and project design, all patients underwent the relay puncture procedure, and technical success was achieved in all occlusive arteries (Figs. 3 and 4). Furthermore, no complication occurred during the treatment. A total of 87 stents (median: 3) were implanted in 21 patients. Perioperative complications included hematoma (one patient) at the brachial artery puncture site and heart failure (one patient) before discharge. These two patients were successfully treated and discharged after

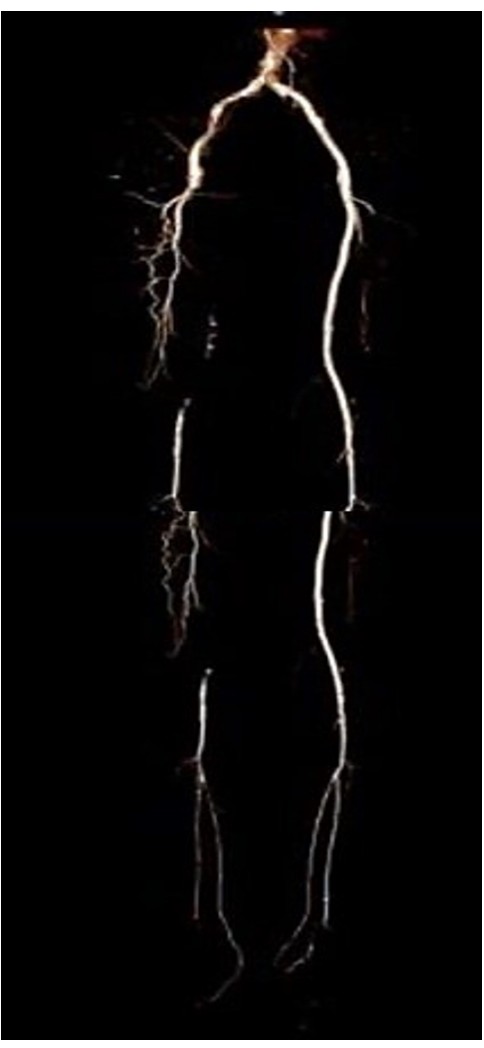

**Figure 1 The CTA image for patient 1 before the operation.** This image shows that the right SFA was totally occluded, and cover stents were implanted in the entire left CIA to SFA, which was unobstructed. The left femoral artery could not be punctured due to cover stents.

treatment. Moreover, the postoperative ABI significantly increased ($0.33 \pm 0.18$ vs. $0.75 \pm 0.21$, $P < 0.05$). Correspondingly, the ischemia symptoms were immediately alleviated after the treatment, as evaluated by RCC (Table 1).

### Long-term outcomes

All patients were followed up for 4–28 months (mean: 16.6 months). During the follow-up period, no serious complications were observed. Color Doppler ultrasonography revealed that the 1-year primary patency rate was 71.43% (15/21). The changes in ABI of these patients during the follow up period are summarized in Table 1.

## DISCUSSION

Since the SAFARI technique was invented in clinic, an increasing number of patients who suffered from lower extremity arterial disease have been successfully treated with

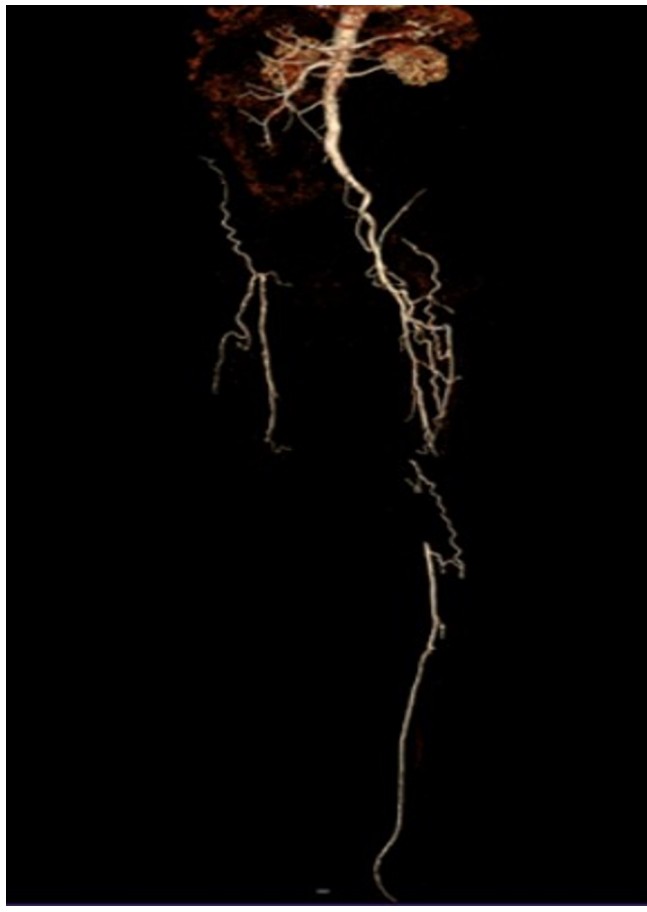

**Figure 2 CTA image for patient 2 before the operation.** The image shows that the left SFA was totally occluded. The right leg had a high position amputation 1 year ago, and the right iliac artery was also occluded. The right femoral artery could not be punctured due to the amputation, and the patient only required recanalization due to economic issues.

endovascular repair (*Hendricks & Sabri, 2014*). The SAFARI technique has also broadened the indications for endovascular therapy in patients with lower extremity arterial disease. However, in some complex situations, lesions could not be merely cured by endovascular treatment, even with the SAFARI technique. The patients presented in the present study had a complicated pathophysiology in diseased lower extremities, and this could not be treated by merely traditional endovascular methods. Therefore, the two position's puncture method was applied to complete the endovascular treatment for these patients. The investigators named this method the relay puncture technique. The present study suggests that the relay puncture technique has appreciable safety and efficacy.

## Relay puncture technique

Patients in the present study did not have access on the contralateral femoral artery due to different reasons. The hybrid operation, which is a combined endovascular and surgical technique, was used as an alternative to laparotomy to correct inflow lesions in patients with multilevel arterial diseases (*Madera et al., 1997*), and this has achieved

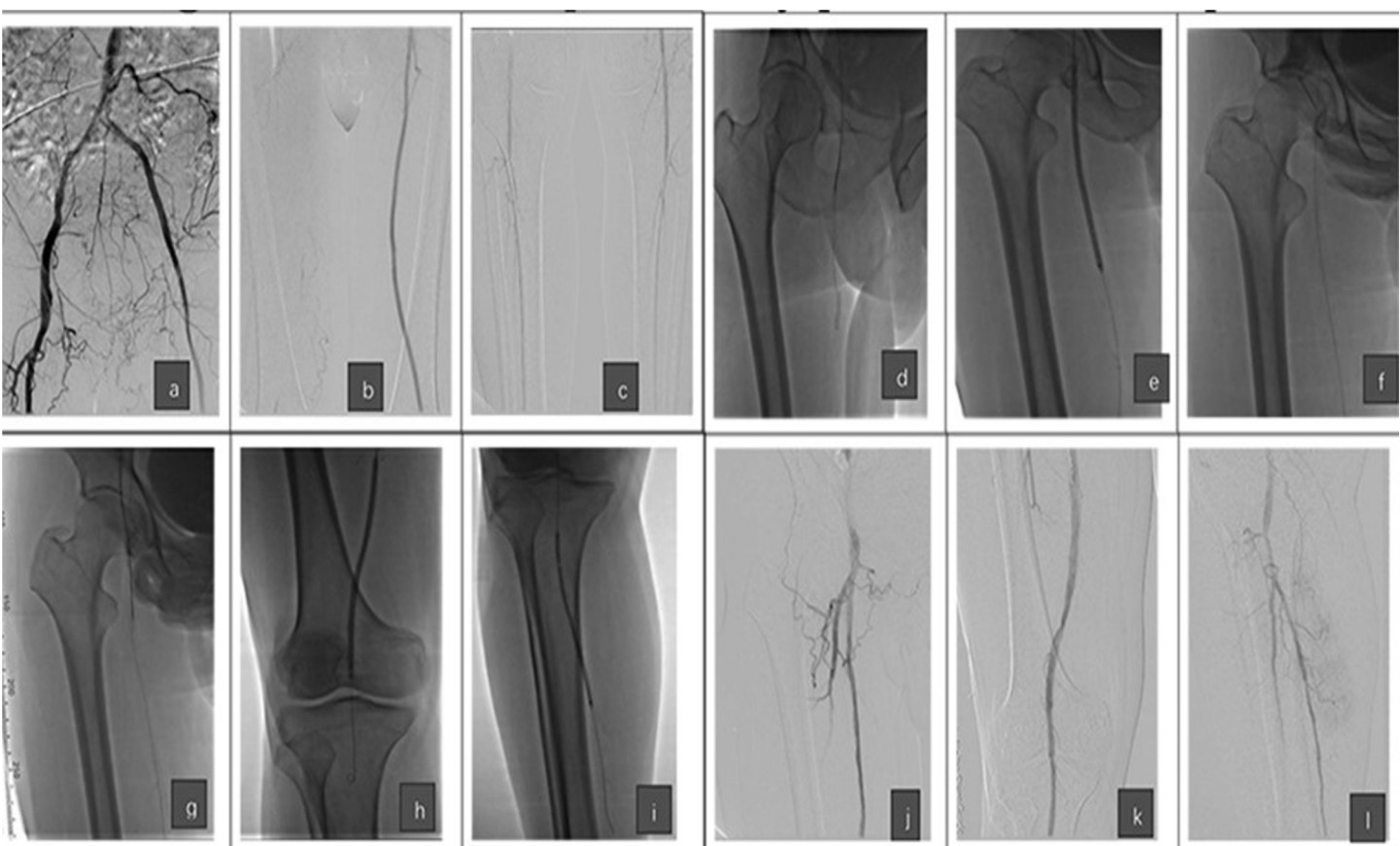

**Figure 3 Images showing the technical steps of the relay puncture technique performed for patient 1.** (A–C) Images of the angiography before treatment show the occlusion of the right SFA and the cover stents of the left side. (D and E) The images show that the recanalization was performed to the proximal part of right SFA through the left brachial access, and the PTA was first performed. (F and G) The images show that the relay puncture technique (antegrade puncture) was performed to the right femoral artery under fluoroscopy and the guidance of the guide wire, which was just recanalized. (H and I) The images show that the recanalization was further performed to the distal part of the right SFA and the arteries below the knee after the relay puncture. (J–L) The images show the angiography after endovascular treatment and the removal of the sheath in the right femoral artery. Note that the recanalization was successful without contrast medium overflow.

appreciable patency rates and technical success rates (*Mousa et al., 2010*). However, the hybrid surgery exhibited several disadvantages, including the requirement of highly skilled surgeons (there are limited numbers of vascular surgeons), the need for an operating room (the cost of a hybrid operating room is high and cannot be sufficiently provided), and the potentially increased risk of complications and longer recovery period (including the risk correlated to further anesthesia) (*Kaneko & Davidson, 2014*). For example, one recent meta-analysis that compared the outcomes of the endovascular and open bypass treatment for TASC C–D aortoiliac occlusive disease reported more complications and greater 30-day mortality with open bypass (*Tsai et al., 2015*).

In the present study, the relay puncture technique, which is a new puncture method with different approaches, was proposed to handle these complex vascular lesions. The word "relay" was used in this technique, because the antegrade puncture on the femoral artery after the brachial artery puncture was just like a relay race. This technique has been applied in 21 patients, and the short- and long-term outcomes appeared to be

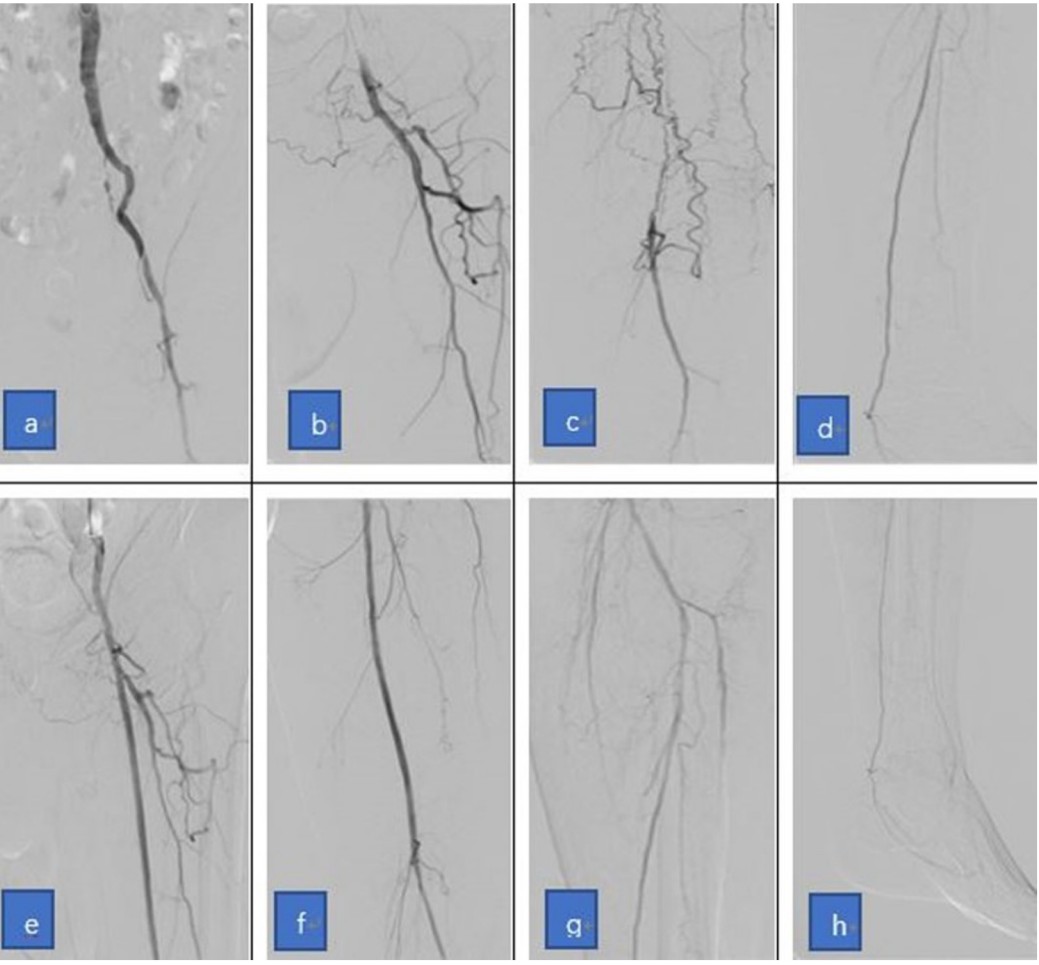

**Figure 4** **The images show the relay puncture technique used for patient 2.** (A–D) The angiography before treatment shows the occlusion of the left SFA and the right iliac to the femoral artery, and that the right femoral artery could not be punctured due to the amputation. (E–H) Similar to Fig. 3, the angiography shows that the left SFA was successfully recanalized after the application of the relay puncture technique.                               

satisfactory, as evidenced by the findings that the technical success rate was 100%, and that the primary long-term patency was 71.43%, which were similar to those previously reported in studies on patients after endovascular treatment (*Klein & Ross, 2016*).

For patients in the present study, there was no puncture site available on the lower extremity arteries. Thus, the investigators opted to puncture the left brachial artery first. However, all endovascular devices failed to reach the lesions at the distal part of the femoral and popliteal artery. The SAFARI technique should be able to establish a guidewire line through the whole lesion. However, the size of both the stent delivery catheter and balloon catheter were too big to be pulled into the arteries below the knee. This was the same with the atherectomy catheter and drug-coated balloon. Therefore, the SAFARI technique could not solve the problem of these patients in the present study, who had long lesions that involved the popliteal artery, or even those below the knee. Directly puncturing the stents on the contralateral femoral artery
**Table 1  Clinical symptom improvements of patients within 1 year in the present study.**

| Patient number | Before treatment | | Immediately after treatment | | 3 months after treatment | | 6 months after treatment | | 12 months after treatment | |
|---|---|---|---|---|---|---|---|---|---|---|
| | RCC | ABI | RCC | ABI | RCC | ABI | RCC | ABI | RCC | ABI |
| 1 | 6 | 0 | 0 | 0.55 | 0 | 0.63 | 1 | 0.58 | 4 | 0.42 |
| 2 | 4 | 0.51 | 0 | 0.81 | 0 | 0.79 | 0 | 0.83 | 0 | 0.65 |
| 3 | 4 | 0.48 | 0 | 0.93 | 0 | 0.90 | 0 | 0.88 | 0 | 0.92 |
| 4 | 5 | 0.22 | 0 | 0.68 | 0 | 0.71 | 0 | 0.68 | 0 | 0.77 |
| 5 | 4 | 0.36 | 0 | 0.99 | 0 | 1.03 | 0 | 0.98 | 0 | 0.95 |
| 6 | 6 | 0 | 1 | 0.70 | 1 | 0.82 | 1 | 0.79 | 4 | 0.58 |
| 7 | 3 | 0.60 | 0 | 1.08 | 0 | 1.05 | 0 | 1.02 | 0 | 0.98 |
| 8 | 4 | 0.41 | 0 | 0.93 | 0 | 0.93 | 0 | 0.88 | 0 | 0.80 |
| 9 | 5 | 0.45 | 0 | 1.12 | 1 | 1.05 | 0 | 1.02 | 0 | 0.95 |
| 10 | 5 | 0.36 | 0 | 0.81 | 0 | 0.88 | 0 | 0.90 | 0 | 0.89 |
| 11 | 4 | 0.72 | 0 | 1.10 | 0 | 1.00 | 0 | 0.98 | 0 | 0.96 |
| 12 | 6 | 0 | 1 | 0.69 | 1 | 0.73 | 1 | 0.63 | 5 | 0.36 |
| 13 | 4 | 0.58 | 0 | 1.03 | 0 | 1.05 | 0 | 0.95 | 0 | 0.90 |
| 14 | 4 | 0.35 | 0 | 0.95 | 0 | 0.99 | 0 | 0.86 | 0 | 0.79 |
| 15 | 5 | 0.33 | 0 | 0.88 | 0 | 0.80 | 0 | 0.79 | 0 | 0.66 |
| 16 | 6 | 0.22 | 0 | 0.93 | 0 | 0.99 | 0 | 0.85 | 4 | 0.45 |
| 17 | 5 | 0.17 | 0 | 0.99 | 0 | 0.97 | 0 | 0.91 | 1 | 0.86 |
| 18 | 4 | 0.32 | 0 | 1.02 | 0 | 1.00 | 0 | 0.89 | 0 | 0.92 |
| 19 | 6 | 0 | 1 | 0.76 | 1 | 0.77 | 1 | 0.65 | 3 | 0.53 |
| 20 | 3 | 0.45 | 0 | 1.10 | 0 | 1.01 | 0 | 0.93 | 0 | 0.93 |
| 21 | 6 | 0.22 | 0 | 0.93 | 0 | 0.99 | 0 | 0.85 | 4 | 0.45 |
| Median/Mean | 5 | $0.33 \pm 0.18$ | 0 | $0.90 \pm 0.16^*$ | 0 | $0.91 \pm 0.13^*$ | 0 | $0.85 \pm 0.16^*$ | 0 | $0.75 \pm 0.21^*$ |

Notes:
RCC, Rutherford Clinical Classification. Least significant difference $t$-test.
* $P < 0.05$ (both compared with ABI/RCC before treatment). The median value is for RCC, and the mean value is for ABI, which were presented as mean ± standard deviation.

should have been a solution. However, that technology is not mature yet, and few studies have reported good long-term outcomes (*Jongkind et al., 2010*). Hence, directly puncturing the contralateral femoral artery appeared as the only option (*Palena & Manzi, 2013*). Another approach that might solve the above-mentioned issue for these patients was to insert a large sheath with a drug-coated balloon or debulking devices through the stent. This approach may damage the stent structure, subsequently accelerating the occurrence of stent rupture and/or restenosis within the stent. In addition, this technique cannot be used for patients with cover stents. Furthermore, in some cases, stents were implanted into the aortic artery in some aorticiliac lesions, and it was difficult for the devices to cross over the iliac branches. Most importantly, all patients in the present study had lesions that involved the area of the upper inguinal, which could not be directly antegrade punctured through the femoral artery. Based on the above-mentioned analysis, the investigators decided to use the new relay puncture technique.

### Puncture of the brachial artery

The antegrade approach using the trans-brachial access with long sheaths and dedicated catheters appeared to be the choice for aortoiliac occlusions with or without extension to the femoral-popliteal arterials. This approach offers better push ability, allows access to both iliac arteries, and enables the reentry at the femoral artery to be managed more easily and safely, when compared with the retrograde approach (*Millon et al., 2015*). Although this involved some risks and technical challenges, the investigators continued to choose this approach for patients in the present study, because they had no other puncture channels. It turned out that this antegrade approach was successfully performed in the present study.

### Antegrade puncture of the superficial femoral artery

At present, performing an antegrade puncture has become a routine in endovascular treatment for patients suffering from infra-inguinal lower extremity arterial disease (*Nice et al., 2003*). Recent studies have shown that antegrade access *via* the superficial femoral artery (SFA) has practical advantage over classical access via the CFA, in terms of shorter access time, quicker fluoroscopy time, and higher success rate (*Gutzeit et al., 2011*; *Kweon et al., 2012*). In the present study, before the SFA was punctured, the guidewire recanalized the vessel and reached the target. Therefore, under fluoroscopy, the antegrade procedure to puncture the guidewire was safe with fewer complications. However, the number of attempts of reentry procedures with the guidewire should be limited to avoid extending the dissection. Regardless of whether it was successful or not, the guidewire guided the antegrade puncture performed to the SFA. In addition, this approach should be performed at the same axis, while the former guidewire should be performed in the lumen, in order to allow these two guide wires to link together. Basically, the first wire created an access, while the second wire took over the treatment of femoral-popliteal lesions, as well as arterial lesions below the knee. The relay approach not only recanalized these infra-inguinal diseases, but also overcame the short-shaft, further increasing the pushing ability and maneuverability of the wire, and the probability of reentry, especially near the popliteal artery. Moreover, the latter wire may be used to reconstruct the arterials below the knee.

## Application of puncture closure devices

Endovascular treatment has been shown to be associated with several complications. For instance, the incidence of hematoma can reach up to 6% for patients undergoing brachial puncture (*Alvarez-Tostado et al., 2009*), and the incidence of complications following the antegrade puncture of SFA can reach up to 9% (*Mlekusch et al., 2008*). However, the use of closure devices exhibited a significant tendency to decrease these complications (*Gutzeit et al., 2014*). Consistent with these reports, in the present study, the median compression time was only 5 min (3–10 min), even in the presence of heparin, which was shorter than the traditional compression time for hemostasis. Furthermore, using the Exoseal device to close the antegrade puncture hole on the femoral artery, endovascular therapy could be quickly

continued without bleeding in the present study. A variety of arterial closure devices have been introduced with the aim of reducing bleeding risk. However, all were indicated for femoral use only (*Hon et al., 2009*). In the present study, Exoseal was used for brachial artery puncture site closure, which is a safe technique with an acceptable rate of complications (6%). Thus, no clinical intervention was required.

## LIMITATIONS OF THE PRESENT STUDY

The present study was a retrospective study with a limited sample size. Hence, the present findings need to be further corroborated in large cohort studies in the future. In addition, the procedures for the relay puncture technique remains complex, and needs to be performed by an experienced doctor.

## CONCLUSION

The present study described the modalities of the "relay puncture" technique. The preliminary results of the present study revealed that this may be a feasible technique in the hands of experienced and skilled operators for the treatment of patients with lower extremity arterial diseases, when the contralateral femoral artery cannot be punctured and the length of the devices is too short to treat the distal lesion of the femoral and popliteal artery through the brachial artery approach.

### Funding

This research was supported by "the Fundamental Research Funds for the Central Universities" of China (No. 21617318). The funders had no role in study design, data collection and analysis, decision to publish, or preparation of the manuscript.

### Grant Disclosure

The following grant information was disclosed by the authors:
Fundamental Research Funds for the Central Universities: 21617318.

### Competing Interests

The authors declare that they have no competing interests.

### Author Contributions

- Chengzhi Li conceived and designed the experiments, performed the experiments, contributed reagents/materials/analysis tools, prepared figures and/or tables, authored or reviewed drafts of the paper, approved the final draft.
- Huimin You performed the experiments, analyzed the data, contributed reagents/materials/analysis tools.
- Hong Zhang performed the experiments.
- Yulong Liu performed the experiments.
- Wanghai Li performed the experiments, analyzed the data.
- Xiaobai Wang authored or reviewed drafts of the paper.

- Yan Zhang conceived and designed the experiments, performed the experiments, authored or reviewed drafts of the paper, approved the final draft.

## Human Ethics

The following information was supplied relating to ethical approvals (i.e., approving body and any reference numbers):

The experimental protocol of the present study complied with the principles outlined in the Declaration of Helsinki and was approved by the Ethics Committee of our institution (No. jnu3287).

## Data Availability

The raw data are available in the Supplementary File (patients' history), Table 1 (Rutherford clinical classification and ABI), and the 'Materials and Methods' section (TASC classifications).

## Supplemental Information

Supplemental information for this article can be found online at http://dx.doi.org/10.7717/peerj.6345#supplemental-information.

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
