# Peer review of "Application of relay puncture technique in treating patients with complicated lower extremity arterial diseases"

_PeerJ, doi:10.7717/peerj.6345_

## Round 0.1 · original submission · Major Revisions

AS noted by the reviewers, the paper would benefit from some input from a native English speaker. The TASC classification of the treated lesions would be helpful.

Reviewer 1 ·

Basic reporting

Mostly clear and professional language but multiple small errors, including grammar and spelling mistakes (for example, branchial frequently used instead of brachial), occasional errors in syntax. Some sentences too long and unclear. The paper should be reviewed/edited by a native English speaker prior to submission to an English language journal.
More context and background should be provided. In particular the authors should defend further why they think the relay technique is superior to a hybrid approach.

I think reference 9 is not relevant as a justification for the relay approach as this paper deals with aortoiliac occlusive disease, ie comparing aortofem bypass to an endovascular approach. Reference 8 is similar. I think the authors should quote evidence that a hybrid approach to the CFA (ie CFA endarterectomy and endovasculat treatment distally) is inferior. There are multiple papers in the literature showing that a hybrid approach is safe and feasible

The article was professionally structured. Figures 1 and 2 are poorly reproduced and difficult to interpret. Raw data provided although clinical data would be more relevant in addition to ABI

Experimental design

Design - a retrospective cohort study with small numbers. Inclusion criteria not well-defined. The degree of disease should be classified, ie TASC criteria. RCC provided. How many patients in total in the institution were treated during the study period? What was their outcome? What was the experience of the treating radiologists (i.e. how many previous cases etc)
Aim of paper – research question
The authors define what they see as the research question, but I am not convinced it is relevant. The authors criticise hybrid procedures as they require vascular surgeons. The paper is written by interventional radiologists. In many countries, open and endovascular procedures are carried out by vascular surgeons not radiologists , and usually either in an operating theatre or a hybrid room, therefore performing hybrid procedures is straightforward and avoids puncturing the brachial artery unnecessarily. Even in the UK for example, where interventional radiologists perform endovascular procedures, it is common to perform hybrid cases where the vascular team treat the common femoral artery via an open approach and the interventional radiologists treat disease distally with access established. The relay puncture technique therefore would have less relevance.

Methods
The authors mention that in seven cases, patients had stents implanted in the femoral artery which could then not be punctured. Do they mean the common femoral (CFA)? It is unusual to stent the CFA.
Methods otherwise well described
Can they elaborate on the SAFARI technique.
It is unclear which lower extremity artery was punctured for access. In the discussion section they refer to accessing via the SFA. This is not common practice. Can this be clarified.

Validity of the findings

The relay puncture technique was found to be feasible and safe. the authors describe good technical success rates and low complication rates. The authors report that 87 stents were used in 21 patients. This seems high given that they were not treating iliac lesions. Did primary angioplasty fail in all the patients? And can the authors specify which arterial segments were stented? Do they have a policy of primary stenting?
The 1 year follow-up is quite short. Did the 6 patients with occluded stents at 1 year require treatment? And which stents occluded?
The authors used a closure device for the brachial artery. Most surgeons would not favour this approach.

Additional comments

Overall a small retrospective study with good results for this technique albeit only short term results quoted. My main issues are with applicability of this technique as I don't feel it is superior to current (i.e. hybrid) approaches although may have a role in some cases

Reviewer 2 ·

Basic reporting

The article must conform to professional standards of courtesy and expression.

Literature references, sufficient field background/context provided.

Professional article structure, figs, tables. Raw data shared.

Self-contained with relevant results to hypotheses.

Experimental design

Original primary research within Aims and Scope of the journal.
Research question well defined, relevant & meaningful. It is stated how research fills an identified knowledge gap.
Rigorous investigation performed to a high technical & ethical standard.
Methods described with sufficient detail & information to replicate.

Validity of the findings

Impact and novelty not assessed. Negative/inconclusive results accepted. Meaningful replication encouraged where rationale & benefit to literature is clearly stated.
Data is robust, statistically sound, & controlled.
Conclusion are well stated, linked to original research question & limited to supporting results.
Speculation is welcome.

Additional comments

The problems statement, design, and methodology are clear and well written. The results section is well-written, focused. The writing needs to be improved.

Reviewer 3 ·

Basic reporting

Just note two errors line 115 and line224: Branchial instead of brachial.

Introduction, figures:OK

Experimental design

Aims of the study are: to report efficacy and safety of a new endovascular technique and the METHODOLOGY seems crucial.

In which population??
This is a RETROSPECTIVE study, and we need to know how the patients have been selected: are they consecutive patients? How many procedures have been made by the unit during the study period?

The functionnal status is described but it should be precised whereas the patients present a critical chronic ischemia or acute ischemia.

The arterial lesions should be more precisely described: topography, length, degree of obstruction, thrombosis... The TASK classification may be useful.

More precisions about the inclusion in the procedure: was surgery ever contreindicated or impossible? More generally, does the procedure always fits with the scientific recommendations?

Choosing a technique because of a "cost problem" may be questionable in terms of ethical concerns. (line 129 130)

Validity of the findings

The retrospective nature of the study and the small size of the population do not allow any extrapollation of the results in order to change our practice.
I appreciate your paragraph about the limitations of the study. Don't you think it should be take place BEFORE the conclusions?

Additional comments

The technique described by the authors seems interesting but its value (safety) and its indications have to be re evaluated after multicentric and comparative studies.
By now the present study only allows to say it is a feasible technique in the hands of an experienced and skilled equipe.

---

## Round 0.2 · Minor Revisions

The manuscript is much improved. Reviewer 3 has a number of suggestions on the attached annotated file. Please address these (they are all minor) and resubmit.

Reviewer 2 ·

Basic reporting

no comment

Experimental design

no comment

Validity of the findings

no comment

Additional comments

The problems statement, design, and methodology are clear and well written. The results section is well-written, focused. Authors' responses are clear and point-to-point.

Reviewer 3 ·

Basic reporting

I appreciate the corrections done by the authors.
English language remains somehow crtical and I made some suggestions in the pdf attached file

Experimental design

Poulation description and methodology have been corrected

Validity of the findings

It is still impossible to extrapollate this data to our daily practice.
The small size of the population ant the retrospective character still reperesent the weakness of this study.
I made some suggestions to the authors , specially in the conclusions

Additional comments

in my point of view, the interest of the paper is to describe this"new"technique, to defend its feasability on the basis of this preliminary data in order to arouse future larger, multicentric and comparative trials.

Annotated reviews are not available for download in order to protect the identity of reviewers who chose to remain anonymous.

---

## Round 0.3 · accepted · Accept

Thank you for making those final changes.